# A Smartphone Healthcare Application, *CALO mama Plus*, to Promote Weight Loss: A Randomized Controlled Trial

**DOI:** 10.3390/nu14214608

**Published:** 2022-11-02

**Authors:** Yoshio Nakata, Hiroyuki Sasai, Masahiko Gosho, Hiroyuki Kobayashi, Yutong Shi, Tomohiro Ohigashi, Shinichiro Mizuno, Chiaki Murayama, Satomi Kobayashi, Yuki Sasaki

**Affiliations:** 1Faculty of Health and Sport Sciences, University of Tsukuba, 1-1-1 Tennodai, Tsukuba 305-8577, Japan; 2Research Team for Promoting Independence and Mental Health, Tokyo Metropolitan Institute of Gerontology, Itabashi-ku, Tokyo 173-0015, Japan; 3Department of Biostatistics, Faculty of Medicine, University of Tsukuba, 1-1-1 Tennodai, Tsukuba 305-8575, Japan; 4Department of Internal Medicine, Mito Kyodo General Hospital, University of Tsukuba, 3-2-7 Miyamachi, Mito 310-0015, Japan; 5Graduate School of Comprehensive Human Sciences, University of Tsukuba, 1-1-1 Tennodai, Tsukuba 305-8577, Japan; 6Link & Communication Inc., Chiyoda-ku, Tokyo 102-0094, Japan

**Keywords:** healthcare application, artificial intelligence, behavior change, dietary intake, exercise, mood, sleep quality, *CALO mama Plus*, overweight, weight loss

## Abstract

Mobile applications are increasingly used in healthcare. We have developed a smartphone healthcare application, *CALO mama Plus*, that can register daily diet, exercise, mood, and sleep quality, calculate dietary intake, and provide advice using artificial intelligence technology. This 3-month randomized controlled trial tested the hypothesis that *CALO mama Plus* could promote body weight reduction in Japanese adults with overweight or obesity. We recruited office workers as participants. The key eligibility criteria were an age of 20–65 years and a body mass index of 23–40 kg/m^2^. The primary outcome was body weight change over 3 months. We enrolled 141 participants and randomly assigned them to the intervention (*n* = 72) and control (*n* = 69) groups. The intervention group used *CALO mama Plus*, and the control group did not receive any intervention. The change in body weight was −2.4 ± 4.0 kg and −0.7 ± 3.3 kg in the intervention and control groups, respectively. An analysis of covariance adjusted for related variables showed a significant between-group difference in body weight change (−1.60 kg; 95% confidence interval −2.83 to −0.38; *p* = 0.011). The present study suggests that *CALO mama Plus* effectively promotes weight loss.

## 1. Introduction

An unhealthy diet and physical inactivity are risk factors for all-cause, cancer, and cardiovascular mortality [1]. These unhealthy lifestyles contribute to overweight and obesity, which are risk factors for cardiovascular diseases, diabetes, musculoskeletal disorders, and some types of cancer [2]. Therefore, behavior change is imperative to prevent noncommunicable diseases.

Behavior change is essential in the treatment of obesity. Guidelines for the management of overweight and obesity in adults showed behavior therapy as one of the principal components, which included the regular self-monitoring of food intake, physical activity (PA), and weight [3]. The Diabetes Prevention Program, one of the most successful lifestyle interventions, requires individual case managers and frequent contact with participants as key features [4]. As the involvement of healthcare professionals is costly, using healthcare applications (app) is a potential alternative [5]. Indeed, many healthcare apps were reported to be effective in improving diet and PA [6,7]. The scalability and sustainability of healthcare apps seemed to be the potential reasons for their effectiveness [8]. Rapid progress in artificial intelligence (AI) technology may further enhance the effectiveness of these apps [9,10].

We developed a smartphone healthcare app called *CALO mama Plus* (Link & Communication Inc., Tokyo, Japan) [11]. Users of *CALO mama Plus* can register their daily diet, exercise, mood, and sleep quality. The app has two main features. First, lifestyle information can be easily managed with a single app. In the case of recording meals, nutrients are assessed using image-recognition technology. Second, AI instantly provides advice based on the diet, exercise, sleep, mood, and weight recorded by users. There are approximately 200 million patterns of advice from AI that can motivate and help users to change their behavior. These features provide benefits not only to users but also to diet program providers. In conventional health programs, even web tools, nutritionists manually calculate the nutritional value by looking at meal images, or experts analyze users’ data and provide individual advice [12,13,14]. Since *CALO mama Plus* performs all of these functions automatically, the financial and personnel burden of diet program providers is dramatically reduced. Thus, *CALO mama Plus* is a promising tool that offers users a much easier recording method and more efficient feedback. *CALO mama Plus* was marketed, and there was a validation study of nutrient and food group prediction by AI [11] and an observational study among the app users, where they improved diet quality during the coronavirus disease 2019 (COVID-19) pandemic [15]. However, we did not test the effectiveness of this app for weight loss using a sophisticated research design.

The present randomized controlled trial (RCT) tested the hypothesis that *CALO mama Plus* could promote body weight reduction in Japanese adults with overweight or obesity. This confirmatory trial could help disseminate and implement the *CALO mama Plus* installation, an easy, low-cost, and effective strategy to enhance individual health.

## 2. Materials and Methods

### 2.1. Study Design

The present 3-month RCT aimed to examine the effect of a smartphone healthcare app, *CALO mama Plus*, on weight loss among Japanese adults with overweight or obesity. The inclusion criteria were as follows: (1) age 20–65 years, (2) body mass index (BMI) 23–40 kg/m^2^, (3) those who could install the healthcare app on their smartphones, and (4) office workers who understood the purpose and content of the study and provided written informed consent. The exclusion criteria were as follows: (1) a history of heart or cerebrovascular disease, (2) currently pregnant or desiring pregnancy during the study, (3) exhibited significant changes in body weight during the last 6 months, (4) those whose cohabiting family members participated in this study, and (5) those who were judged as unsuitable for the study by the principal investigator for other reasons. At a prior sample size calculation, we assumed the effect size (Cohen’s d) of this intervention as 0.5, the changes in body weight as 2.5 kg and 0.5 kg in the intervention and control groups, respectively, and the standard deviation as 4.0 kg. With these assumptions at 5% of the significance level and 80% of statistical power, the calculated sample size was 64 per group and 128 in total. As we anticipated a 15% dropout rate after the invitation for the screening test, we set the target sample size to 150.

The ethics review board of the Faculty of Health and Sport Sciences at the University of Tsukuba approved the study protocol (approval number: Tai 019-127) on 21 January 2020. After the COVID-19 pandemic, we decided to conduct all the sessions online. The revised protocol was approved on 1 October 2020 and registered at the University Hospital Medical Information Network (UMIN) Clinical Trial Registry (UMIN000042072) on 10 October 2020. This study followed the Consolidated Standards of Reporting Trials 2010 guidelines [16].

### 2.2. Participants

We recruited participants from four cooperative companies, mainly in the insurance, food, or pharmaceutical industries, and one volunteer database associated with a contract research organization in Tokyo, Japan. We held the first introductory session on the video meeting platform Zoom (San Jose, CA, USA), or a video of the session on YouTube. We requested a screening test, including an online baseline questionnaire and self-administered measurement. A total of 230 adults indicated their interest in participating in the study. We invited 199 adults who potentially satisfied the eligibility criteria for the introductory session. Of these, 176 participated in the session, 168 gave informed consent, 156 replied to the screening test, and 141 were eligible for the present study. All participants provided written informed consent before the eligibility assessment. We prepared some types of rewards comparable to JPY 20,000 (money, gift card, healthcare device, or sporting goods) and offered either one to the participants. The type and timing (before or after the trial) of the rewards depended on the cooperative companies or the contract research organization.

### 2.3. Randomization

Participants who met the eligibility criteria were randomized into the intervention and control groups. A biostatistician prepared a random number sequence. The allocation ratio was 1:1, and the stratification factors were sex and the timing of reward provision (before or after the trial). A study staff member who did not contact the participants maintained the randomization table, which was opened after the final dataset was checked and fixed.

### 2.4. Interventions

The intervention started on 1 December 2020 and ended on 22 February 2021. During this period, the number of people infected with COVID-19 increased rapidly in Japan. On 7 January 2021, the government declared a state of emergency, the second since the initial declaration on 7 April 2020. The declaration ended on 18 March 2021.

The participants in the control group did not receive any interventions. Therefore, we requested them to continue their current lifestyle and not use any dietary apps. In addition, we ensured their chances of using *CALO mama Plus* for 3 months after the trial.

The participants in the intervention group used the *CALO mama Plus* smartphone healthcare app. We requested them to install the app on their smartphones, register their daily weight, diet, exercise, mood, and sleep quality, and live according to the advice from the app. *CALO mama Plus* could show their lifestyle changes graphically, evaluate their lifestyle, and indicate some tips for improvement. The details of *CALO mama Plus* are shown in the subsequent subsections and Figure 1. The study staff checked their data-input frequently and e-mailed them once a week to encourage input if they had not registered their lifestyle information. The criteria for sending the e-mail were inputting less than 4 days per week of diet information or less than once per week of weight, mood, or sleep quality. If a participant received the notice for three consecutive weeks, the study staff called the participants to encourage them.

#### 2.4.1. Setting

*CALO mama Plus* has multiple courses that address various health issues (e.g., weight loss, muscle gain, and weight maintenance). We instructed the participants to select a weight-loss course and input their age, sex, height, and weight. Target values for energy intake, nutrients, and PA were automatically determined from the selected courses and their baseline information, including BMI. The target values were calculated based on clinical practice guidelines and scientific evidence, such as the dietary reference intakes for the Japanese [17].

#### 2.4.2. Conversion of Users’ Input Data

Users record items of every meal in the app, and AI supports accurate recording by detecting photos of the meals taken by users [11,15]. The app has approximately 150,000 foods, including fresh food, ready meals, and commercial products, by brand name and menu selections from approximately 450 stores and restaurants. Therefore, the app identifies and records the names of restaurants and manufacturers if a user eats processed food outside of their cooking area, at a restaurant, or collects takeaway. In the case of ready-made meals, nutrition intake is calculated by applying the recorded food items to a standard menu, which is predefined based on several recipe books. The input data in the app are converted into numerical values that enable a comparison with the target values.

#### 2.4.3. Feedback to Users

*CALO mama Plus* generates appropriate advice by comparing the converted value with the target value. The weight-loss course prioritizes energy and protein intake. The app generates an applause message when intake is within the standard range. The app generates a warning message when it is over a certain amount of the standard. The app generates an encouraging message when it is below and proposes a tip to take more.

*CALO mama Plus* can generate approximately 200 million advice patterns from AI. For example, if the energy intake is an appropriate amount, “Great! Keep it up!”; if the exercise amount is not enough, “Why don’t you set a goal to exercise once in 3 days? Let’s start tomorrow!”; and if the weight has been steadily decreasing, “You’ve been losing weight at a great pace for the past month!” or if the weight has not been steadily decreasing, “By 5 more kg until your target weight! No need to rush!”.

### 2.5. Measurements

All study outcomes were assessed at baseline and at 3 months in a self-administered manner to avoid the risk of COVID-19 infection. We requested the participants to measure their weight, collect blood, wear an accelerometer, and respond to the questionnaires. The baseline measurements were implemented as a screening test on 4–11 November 2020. The 3-month measurements were taken from 16 February to 5 March 2021. The primary outcome was body weight change over 3 months. The secondary outcomes were changes in triglyceride, high-density lipoprotein cholesterol, low-density lipoprotein cholesterol, hemoglobin A1c, fasting plasma glucose, dietary intake, and PA over 3 months. In addition, the participants were requested to report adverse events during the intervention period.

#### 2.5.1. Basic Characteristics

Using self-service questionnaire software (Questant; Macromill, Tokyo, Japan), participants reported their basic characteristics: age, sex, height, weight, waist circumference, systolic and diastolic blood pressure, medical history, medication use, family medical history, weight control history, current smoking, menopausal status, working status, educational attainment, household income, and living status. Of them, height, waist circumference and blood pressure were requested to be transcribed from their workplace annual health checkups.

#### 2.5.2. Body Weight

A weight scale (HD-665; Tanita, Tokyo, Japan) was sent to each participant to measure their weight. We requested measurements in the morning after an overnight (≥10 h) fast and sent the data via Questant. We calculated BMI as the weight in kilograms divided by the square of the self-reported height in meters.

#### 2.5.3. Blood Biochemistry Measures

The participants were requested to collect blood using a test kit (Smaho de Dock; KDDI, Tokyo, Japan), whose methodology and validation have been reported elsewhere [18]. Briefly, the test kit comprised a tube with a dilution buffer solution, a lancet, a blood-aspiration sponge, a cylinder with a blood cell separation filter, a cap for shutting it tightly, a swab, and a band-aid. According to the instructions, the participants collected approximately 65 μL of blood from their fingertips, isolated diluted plasma by themselves, and mailed the samples to a laboratory.

#### 2.5.4. Dietary Intake

We used the brief self-administered diet history questionnaire (BDHQ), a short version of the self-administered diet history questionnaire previously validated in Japan [19,20]. Details of the BDHQ structure, method of calculating dietary intake, and validity for commonly studied food and nutrient intake have been published elsewhere [19,20]. Briefly, the BDHQ is a 4-page fixed-portion questionnaire that assesses dietary intake based on the reported frequency of consumption of 58 food and beverage items. It takes approximately 15 min to answer. Most food and beverage items were selected from the food list of the diet history questionnaire, as these items are popular in Japan and are listed in the National Health and Nutrition Survey (NHNS) Japan. Standard portion sizes were based on various recipe books for Japanese dishes. 

The BDHQ includes the frequency of consumption of selected foods, usual cooking methods, and general dietary behavior. We mailed the questionnaire to the participants and requested them to fill it out and send it back. The responses to the BDHQ were checked once by the research staff. If any missing or erroneous responses were given to the essential questions, the participants were asked to complete the questions again. We calculated the energy intake in kcal and percent intake of macronutrients (protein, fat, and carbohydrate). The energy intake of participants in the intervention group was also calculated in the first and last 2 weeks using the dietary logs in the app.

#### 2.5.5. Physical Activity

We used a validated triaxial accelerometer (Active style Pro HJA-750C; Omron Healthcare, Kyoto, Japan) that can count steps and estimate the intensity of PA based on metabolic equivalents (METs) from a published algorithm [21,22]. We sent the device to the participants and requested them to wear it on their waist during waking hours, except during water activities or in specific exercises for safety reasons (e.g., contact sports) for 7 days, and send it back after completing the measures. We collected the data in a 60-s period. If there were no acceleration signals for 60 consecutive minutes, the period was defined as “non-wear” [23]. The daily record was valid when the participants wore the device for at least 10 h/day [24]. We excluded those records from the analysis when the valid days were less than 3 days. Finally, we calculated the mean daily step count and time spent in moderate-to-vigorous PA (MVPA; ≥3 METs).

### 2.6. Statistical Analysis

A statistician prepared a statistical analysis plan before fixing the final dataset. All analyses followed a predetermined plan using SAS (version 9.4; SAS Institute, Cary, NC, USA). Statistical significance was set at *p* < 0.05. Participants’ baseline characteristics were shown as mean (standard deviation) for continuous variables or as number (frequency in percentage) for categorical variables. The changes in continuous variables were shown as mean (standard deviation). Our primary analysis followed the intention-to-treat principle without imputation of missing data. An analysis of covariance was used to examine the statistical significance of 3-month changes in primary and secondary outcomes between the groups using sex, the timing of providing rewards, and each baseline value as covariates. The primary outcome was also compared using the Wilcoxon rank sum test as a sensitivity analysis.

## 3. Results

Figure 2 shows the flowchart of the participants. Eligible participants were randomized into the intervention group (*n* = 72) and the control group (*n* = 69). Most participants received rewards after the trial, whereas five (three and two in the intervention and control groups, respectively) received rewards before the trial. The baseline characteristics are presented in Table 1. One participant in the intervention group discontinued the trial due to a busy schedule and the inability to submit data. Therefore, the dataset for the remaining 140 participants was included in the primary analysis. The participants reported no clinically significant adverse events.

During the intervention period, the mean frequencies of inputting information were 0.79 ± 0.25 times per day for their weight, 3.53 ± 0.64 for diet, 0.40 ± 0.49 for exercise, 0.82 ± 0.23 for mood, and 0.84 ± 0.21 for sleep quality. The mean frequency of inputting any information was 0.96 ± 0.12 times per day. The study staff e-mailed or called the low-compliant participants 81 participant-times. Of the 81 follow-ups, 7 were conducted by phone. Of the 72 participants, 31 (43%) were followed up at least once. In addition, we sent a message to the participants in the intervention group six times to communicate caution to input their information on the app, greeting year-end and new-year holidays, and notices of the remaining intervention period.

Table 2 presents the analysis results of the primary and secondary outcomes. The 3-month body weight change in the intervention and control groups was −2.4 kg and −0.7 kg, respectively. The adjusted mean difference was −1.60 kg (95% confidence interval (CI) −2.83 to −0.38). The secondary outcomes tended to improve greater in the intervention group, whereas there were no significant differences between the groups.

Table 3 shows the changes in dietary intake and PA over 3 months. Although the energy intake tended to decrease in both groups, there were no significant differences between the groups. According to the dietary logs of the intervention group, energy intake decreased from 1833 kcal in the first 2 weeks to 1682 kcal in the last 2 weeks. The energy intake calculated by the app significantly decreased (−152.3 ± 304.0 kcal; *p* < 0.001). In addition, PA tended to decrease in both groups. The intervention group showed greater maintenance; however, there were no significant differences between the groups.

## 4. Discussion

The present 3-month RCT demonstrated the effectiveness of the smartphone healthcare app *CALO mama Plus* in promoting body weight reduction in Japanese adults with overweight or obesity. An analysis of covariance adjusted for related variables showed a significant difference in body weight change (−1.60 kg; 95% CI −2.83 to −0.38; *p* = 0.011) between the intervention and control groups. This confirmatory trial could provide evidence to help disseminate and implement the *CALO mama Plus* installation to enhance individual health.

Many healthcare apps have been reported to improve diet and PA [6,7]. A recent systematic review of web-based weight loss interventions found web-based interventions to be more effective than an inactive (wait-list) control group [14]. The meta-analysis showed that the mean difference (95% CI) of weight changes was −2.14 kg (−2.65 to −1.64 kg). The present study demonstrated similar effect sizes. In addition, in the Japanese nationwide interventional program for targeting metabolic syndrome, a body weight reduction of 3%–5% was considered a feasible weight loss target [25,26]. The minimum weight reduction required to improve obesity-related risk factors was reported to be 3% [26]. The mean change in body weight (−2.4 kg) in the intervention group corresponded to a 3.1% change. Although the present study did not demonstrate the effectiveness of blood biochemistry measures, a modest degree of weight loss may enhance individual health.

There were no significant differences in dietary intake and PA over 3 months; however, the effectiveness on body weight change was observed. The validation of the dietary questionnaire BDHQ has been reported [19,20]; however, self-reported energy intake is known to be underestimated [27]. Therefore, the questionnaire might not be able to track dietary changes with high reliability and responsiveness. According to the dietary logs of the intervention group, energy intake significantly decreased from 1833 kcal in the first 2 weeks to 1682 kcal in the last 2 weeks, which was considered a possible cause of weight loss. Regarding PA, both groups decreased step count and MVPA. This observation could be because a nationwide state of emergency was declared due to the COVID-19 spread. The COVID-19 pandemic led national and local governments to suggest mild lockdowns or stay-at-home advisories. Several reports have determined that domestic quarantine introduced a shift in the way of life in the direction of restricted socialization and decreased PA and steps [28,29,30]. However, the intervention group tended to have a greater maintenance of steps (3-month change in the intervention and control groups, −401 ± 3613 and −1271 ± 2391, respectively; *p* = 0.055 for group difference) during the COVID-19 pandemic. This may enhance energy expenditure and contribute to a negative energy balance.

According to the Japanese criteria [31], people with a BMI of ≥25 kg/m^2^ are defined as obese, while the inclusion criteria of the participants’ BMI were 23–40 kg/m^2^ in this study. According to the NHNS Report by the Ministry of Health, Labour and Welfare, the obesity rate (BMI ≥ 25) for Japanese people aged over 20 years is 33.0% for men and 22.3% for women [32]. According to the NCD Risk Factor Collaboration [33], the adult obesity rate (BMI ≥ 30) in Japan is 4.97%, ranking 157th worldwide. These data suggest that Japan’s obesity rate is low on a global scale. In addition, according to the National Institute for Health and Care Excellence guidelines [34], the use of lower BMI thresholds to trigger behaviors to reduce the risk of conditions such as type 2 diabetes has been recommended for adults of Asian family origin. The lower threshold is 23 kg/m^2^, indicating an increased risk. Considering the low prevalence of obesity in Japan and the lower threshold of increasing risk, we designated a BMI of 23 kg/m^2^ to allow for the recruitment of this study.

The mean number of diet registrations was more than three times per day (3.53 ± 0.64) because of the app’s function, which reminds users to register their three meals and snacks. The same function reminds users to register their weight, exercise, mood, and sleep every day, whereas the mean number of exercise registrations was relatively low (0.79, 0.40, 0.82, and 0.84 times per day, respectively). This is because of the low proportion of those who exercise regularly. The average proportion in Japan is 33.4% and 25.1% in men and women, respectively, according to the NHNS [32]. In the intervention group, 30.6% and 33.8% of the participants exercised regularly before and after the intervention, respectively. This app’s function is assumed to assist users in appropriately self-monitoring their daily behaviors and contribute to reducing body weight.

The type of rewards and timing (before or after the trial) were not the same and depended on the cooperative companies or contract research organization. Most participants received gift cards after the trial, whereas five (three and two in the intervention and control groups, respectively) received healthcare devices before the trial. Because this discordance had the potential to affect the statistical analyses, we used the timing of reward provision as one of the covariates.

The major strength of this study was its sophisticated research design. The internal validity of the effectiveness trial is high. As *CALO mama Plus* can be freely downloaded and installed, it is easy to disseminate in healthcare settings.

This study has several limitations. First, all participants were Japanese and more than 70% were men, which limits the generalizability of this study. Second, the study duration was relatively short. Further studies are necessary to examine the long-term effectiveness. Third, the study staff checked the data input frequencies to enhance the adherence of the intervention group. Without this intervention, the effect size could be reduced. Fourth, the energy intake measured by the BDHQ and the physical activity measured by an accelerometer could not fully explain the primary result of the body weight change. Finally, *CALO mama Plus* is a Japanese edition, limiting its dissemination and implementation in other countries.

## 5. Conclusions

The present 3-month RCT demonstrated the effectiveness of *CALO mama Plus* in promoting body weight reduction in Japanese adults with overweight or obesity. Our results suggest that this app can be a cost-effective option for promoting weight loss in the short term and potentially enhancing individual health.

## Figures and Tables

**Figure 1 nutrients-14-04608-f001:**
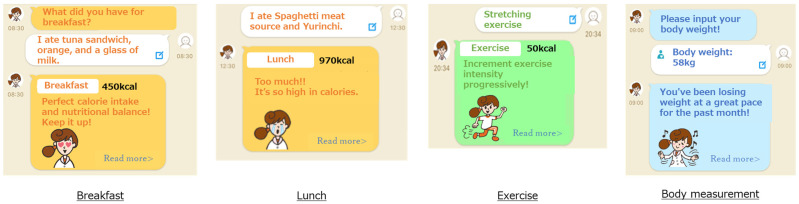
Screenshots of *CALO mama Plus*.

**Figure 2 nutrients-14-04608-f002:**
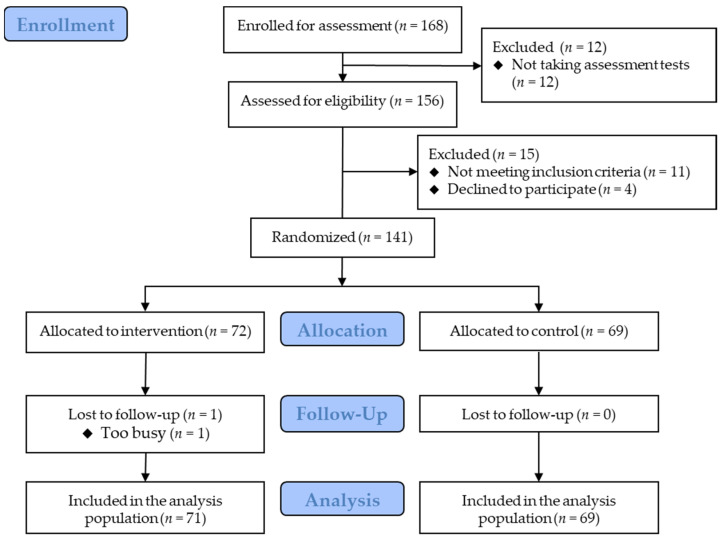
Participants’ flowchart.

**Table 1 nutrients-14-04608-t001:** Participants’ baseline characteristics by the assigned group.

	Intervention (*n* = 72)	Control (*n* = 69)
Age, years	42.3 ± 9.4	44.0 ± 9.1
Sex (men), *n* (%)	53 (74)	51 (74)
Height, cm	167.6 ± 7.5	168.6 ± 8.4
Body mass index, kg/m^2^	27.3 ± 3.3	27.8 ± 3.7
Waist circumference, cm	90.7 ± 8.7	92.2 ± 8.6
Systolic blood pressure, mm Hg	120.3 ± 12.5	126.6 ± 13.9
Diastolic blood pressure, mm Hg	77.2 ± 9.5	80.4 ± 11.6
Medical history, *n* (%)		
Any	27 (38)	28 (41)
Fatty liver	10 (14)	15 (22)
Hypertension	10 (14)	13 (19)
Dyslipidemia	17 (24)	21 (30)
Diabetes	3 (4)	4 (6)
Gout	7 (10)	7 (10)
Medication use, *n* (%)		
Any	20 (28)	8 (12)
Antihypertensive	6 (8)	3 (4)
Lipid-lowering	7 (10)	4 (6)
Hypoglycemic	2 (3)	2 (3)
Antigout	4 (6)	1 (1)
Family medical history, *n* (%)		
Stroke	17 (24)	16 (23)
Heart disease	15 (21)	8 (12)
Hypertension	36 (50)	33 (48)
Dyslipidemia	22 (31)	15 (22)
Diabetes	24 (33)	20 (29)
Weight control history		
Lifetime maximum weight, kg	79.7 ± 10.2	82.8 ± 12.9
Age at maximum weight, years	39.2 ± 10.5	38.7 ± 10.2
Weight at the age of 20 years, kg	63.6 ± 8.8	65.1 ± 11.7
≥3 kg of weight loss in the past, *n* (%)	37 (51)	44 (64)
Current smoking, *n* (%)	7 (10)	10 (14)
Current exercise habit, *n* (%)	22 (31)	21 (30)
Premenopausal women, *n* (% of women)	11 (58)	11 (61)
Working status		
Employed full-time, *n* (%)	64 (89)	60 (87)
Shift or late-night work, *n* (%)	2 (3)	1 (1)
Four-year college graduate or higher, *n* (%)	52 (72)	52 (75)
Household income, *n* (%)		
<3,000,000 JPY	8 (11)	4 (6)
3,000,000 JPY to 5,000,000 JPY	10 (14)	8 (12)
5,000,000 JPY to 7,000,000 JPY	14 (19)	17 (25)
7,000,000 JPY to 10,000,000 JPY	21 (29)	24 (35)
≥10,000,000 JPY	19 (26)	16 (23)
Living alone, *n* (%)	14 (19)	11 (16)
Married, *n* (%)	50 (69)	54 (78)

Data were expressed as mean ± standard deviation or frequency (%).

**Table 2 nutrients-14-04608-t002:** Change in weight and blood biochemistry measures over 3 months.

	Intervention		Control		Adjusted Mean Difference ^1^ (*p*-Value)
	*n*	Mean ± SD	*n*	Mean ± SD	
Weight, kg					
Baseline	72	76.5 ± 9.6	69	79.1 ± 11.3	
Month 3	71	73.8 ± 9.5	69	78.4 ± 12.0	
Change	71	−2.4 ± 4.0	69	−0.7 ± 3.3	−1.60 [−2.83, −0.38] (*p* = 0.011)
Change ^2^ Median [Q1, Q3]	71	−1.3 [−3.2, −0.5]	69	−0.4 [−1.6, 0.9]	(*p* < 0.001)
Triglyceride, mg/dL					
Baseline	72	140.6 ± 99.8	69	161.5 ± 127.4	
Month 3	71	129.7 ± 139.3	67	171.1 ± 142.5	
Change	71	−9.9 ± 87.7	67	8.6 ± 144.8	−23.73 [−61.94, 14.47] (*p* = 0.22)
HDL cholesterol, mg/dL					
Baseline	72	60.9 ± 12.0	69	57.9 ± 12.8	
Month 3	71	63.6 ± 12.8	67	58.4 ± 12.0	
Change	71	2.4 ± 6.1	67	1.4 ± 7.6	1.42 [−0.84, 3.68] (*p* = 0.22)
LDL cholesterol, mg/dL					
Baseline	72	122.3 ± 30.1	69	119.0 ± 30.5	
Month 3	71	118.8 ± 28.3	67	117.9 ± 28.4	
Change	71	−3.2 ± 15.5	67	0.0 ± 17.2	−2.45 [−7.50, 2.61] (*p* = 0.34)
Hemoglobin A1c, %					
Baseline	72	5.6 ± 0.4	69	5.5 ± 0.4	
Month 3	71	5.6 ± 0.4	67	5.6 ± 0.5	
Change	71	0.0 ± 0.1	67	0.1 ± 0.2	−0.04 [−0.08, 0.01] (*p* = 0.15)
Glucose, mg/dL					
Baseline	72	108.2 ± 14.3	69	107.4 ± 15.3	
Month 3	71	106.8 ± 16.0	67	107.7 ± 13.6	
Change	71	−1.5 ± 11.6	67	1.2 ± 11.8	−2.43 [−6.08, 1.23] (*p* = 0.19)

Data were expressed as mean ± standard deviation (SD) or adjusted mean [95% confidence interval] unless specified. ^1^ The model was adjusted for sex, the timing of rewards, and respective baseline values. ^2^ The primary outcome was also compared using the Wilcoxon rank sum test as a sensitivity analysis. HDL, high-density lipoprotein; LDL, low-density lipoprotein; Q1, first quartile; Q3, third quartile.

**Table 3 nutrients-14-04608-t003:** Change in dietary intake by diet history questionnaire and physical activity over 3 months.

	Intervention		Control		Adjusted Mean Difference ^1^ (*p*-Value)
	*n*	Mean ± SD	*n*	Mean ± SD	
Energy intake, kcal/day					
Baseline	72	1866.2 ± 539.9	69	1847.1 ± 655.0	
Month 3	71	1764.2 ± 579.6	69	1715.2 ± 562.4	
Change	71	−85.5 ± 465.4	69	−132.0 ± 513.2	58.7 [−82.5, 199.8] (*p* = 0.41)
Protein intake, %					
Baseline	72	15.3 ± 2.8	69	14.6 ± 2.7	
Month 3	71	15.5 ± 2.4	69	15.2 ± 2.9	
Change	71	0.1 ± 2.9	69	0.7 ± 2.4	−0.18 [−0.94, 0.58] (*p* = 0.63)
Fat intake, %					
Baseline	72	29.1 ± 5.8	69	27.6 ± 7.0	
Month 3	71	29.1 ± 4.8	69	28.1 ± 6.3	
Change	71	−0.1 ± 5.8	69	0.6 ± 6.3	0.18 [−1.45, 1.82] (*p* = 0.82)
Carbohydrate intake, %					
Baseline	72	48.3 ± 9.2	69	47.5 ± 9.6	
Month 3	71	49.0 ± 8.4	69	46.7 ± 10.3	
Change	71	0.8 ± 8.9	69	−0.7 ± 8.0	1.85 [−0.67, 4.36] (*p* = 0.15)
Step count, steps/day					
Baseline	70	8087 ± 3567	69	7751 ± 3145	
Month 3	68	7783 ± 4030	64	6688 ± 3018	
Change	68	−401 ± 3613	64	−1271 ± 2391	922 [−21, 1865] (*p* = 0.055)
MVPA, min/day					
Baseline	70	65.2 ± 39.8	69	60.4 ± 25.8	
Month 3	68	62.7 ± 41.0	64	54.3 ± 23.1	
Change	68	−2.0 ± 33.3	64	−7.5 ± 20.2	6.78 [−1.60, 15.15] (*p* = 0.11)

Data were expressed as mean ± standard deviation (SD) or adjusted mean [95% confidence interval]. ^1^ The model was adjusted for sex, the timing of rewards, and respective baseline values. MVPA, moderate-to-vigorous physical activity.

## Data Availability

The data presented in this study are available upon request from the corresponding author. The data were not publicly available due to privacy concerns.

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
