# Peer review of "A Smartphone Healthcare Application, CALO mama Plus, to Promote Weight Loss: A Randomized Controlled Trial"

_nutrients, 2022, doi:10.3390/nu14214608_

Round 1
Reviewer 1 Report
This study assesses the utility of a healthcare application, CALO mama Plus that uses Artificial Intelligence in weight reduction in a Japanese population over a 3-month period. It is well-written, easy to read and understand. I really enjoyed reading it.
In Western Society overweight is over defined as BMI >25. Is the APP capable of being adjusted to meet this difference? Examples were given of positive reinforcement. What messages are provided when the subject is not meeting targets, or putting on weight?
ITT means that everyone is analysed, yet one person was lost to follow-up, so the abbreviation ITT should be deleted from the paper.
It would be helpful to understand the spread of BMI by group using a histogram. Are there any statistical differences in table 1? If so, then can they be highlighted and if not, then can this be made known via a footnote? For example, baseline weight between groups was almost significant.
The biggest problem that I have is that the data doesn’t form a consistent narrative. In table 3 the drop in energy intake was almost 50% higher in the control group, 132 vs 85.5. On the other hand, the steps per day is almost 3 times that in the control group compared to intervention group. The activity data may be compromised as those participants that had less than 3 valid days of steps are very unlikely to be missing at random so it’s difficult to rely on this information. The primary result is that the loss in weight was more than 3 times greater in the intervention group 2.4 vs. 0.7, but how is this to be understood? It looks like those in the intervention group ate relatively more, exercised more and the exercise won out. The underlying problem with all of this is measurement fidelity. Have any studies been done to assess the reliability of the App? The other problem is the self-measurement. My understanding is that weight loss studies are notorious for participants (in the control group) not being scrupulously honest about their measurements. It may have been prudent to ask them to measure themselves at least once per month. My guess is that the authors need to revisit the data to better understand what is happening.
The upshot is that this the results of this study as they stand have raised several questions that have not been addressed and these results must be interpreted very cautiously.
Author Response
We thank the reviewers for their constructive comments and appreciate the opportunity to revise the manuscript accordingly. Below is a point-by-point response to the concerns raised (reviewer comments are in italics), and in the manuscript, changes have been marked using the track-changes function.
Reviewer #1:
- This study assesses the utility of a healthcare application, CALO mama Plus that uses Artificial Intelligence in weight reduction in a Japanese population over a 3-month period. It is well-written, easy to read and understand. I really enjoyed reading it.
RESPONSE
We appreciate your positive comments.
- In Western Society overweight is over defined as BMI >25. Is the APP capable of being adjusted to meet this difference? Examples were given of positive reinforcement. What messages are provided when the subject is not meeting targets, or putting on weight?
RESPONSE
- ITT means that everyone is analysed, yet one person was lost to follow-up, so the abbreviation ITT should be deleted from the paper.
RESPONSE
In accordance with the comment, we deleted the ITT abbreviation. Please confirm lines 242, 275, and Figure 2.
- It would be helpful to understand the spread of BMI by group using a histogram. Are there any statistical differences in table 1? If so, then can they be highlighted and if not, then can this be made known via a footnote? For example, baseline weight between groups was almost significant.
RESPONSE
In accordance with the comment, we presented the histogram of BMI in Figure R1. The distribution of BMI was not different between the intervention and control groups. In addition, there was no outlier in either group.
Figure R1. Histogram of BMI at baseline (top, intervention group; bottom, control group)
In accordance with the comment, we provided p-values in Table R1. Table R1 shows that systolic blood pressure and medication use statistically differ between the intervention and control groups.
According to the CONSORT statement, hypothesis testing for baseline characteristics in randomized trials is superfluous and can mislead investigators and their readers. Rather, comparisons at baseline should be based on the prognostic strength of the variables measured and the size of any chance imbalances that have occurred. Considering the CONSORT statement, we did not provide the p-values for group comparison in Table 1. In addition, the body weight at baseline did not differ by the two groups (p = 0.15).
Reference:
https://www.consort-statement.org/checklists/view/32--consort-2010/510-baseline-data
Table R1. Baseline characteristics and group comparison
|
|
Intervention (n = 72) |
Control (n = 69) |
p |
|
Age, years |
42.3 ± 9.4 |
44.0 ± 9.1 |
0.28 |
|
Sex (men), n (%) |
53 (74) |
51 (74) |
1.00 |
|
Height, cm |
167.6 ± 7.5 |
168.6 ± 8.4 |
0.45 |
|
Body mass index, kg/m2 |
27.3 ± 3.3 |
27.8 ± 3.7 |
0.34 |
|
Waist circumference, cm |
90.7 ± 8.7 |
92.2 ± 8.6 |
0.32 |
|
Systolic blood pressure, mm Hg |
120.3 ± 12.5 |
126.6 ± 13.9 |
0.006 |
|
Diastolic blood pressure, mm Hg |
77.2 ± 9.5 |
80.4 ± 11.6 |
0.079 |
|
Medical history, n (%) |
|
|
|
|
Any |
27 (38) |
28 (41) |
0.73 |
|
Fatty liver |
10 (14) |
15 (22) |
0.27 |
|
Hypertension |
10 (14) |
13 (19) |
0.50 |
|
Dyslipidemia |
17 (24) |
21 (30) |
0.45 |
|
Diabetes |
3 (4) |
4 (6) |
0.71 |
|
Gout |
7 (10) |
7 (10) |
1.00 |
|
Medication use, n (%) |
|
|
|
|
Any |
20 (28) |
8 (12) |
0.020 |
|
Antihypertensive |
6 (8) |
3 (4) |
0.49 |
|
Lipid-lowering |
7 (10) |
4 (6) |
0.53 |
|
Hypoglycemic |
2 (3) |
2 (3) |
1.00 |
|
Antigout |
4 (6) |
1 (1) |
0.37 |
|
Family medical history, n (%) |
|
|
|
|
Stroke |
17 (24) |
16 (23) |
1.00 |
|
Heart disease |
15 (21) |
8 (12) |
0.17 |
|
Hypertension |
36 (50) |
33 (48) |
0.87 |
|
Dyslipidemia |
22 (31) |
15 (22) |
0.26 |
|
Diabetes |
24 (33) |
20 (29) |
0.59 |
|
Weight control history |
|
|
|
|
Lifetime maximum weight, kg |
79.7 ± 10.2 |
82.8 ± 12.9 |
0.12 |
|
Age at maximum weight, years |
39.2 ± 10.5 |
38.7 ± 10.2 |
0.79 |
|
Weight at the age of 20 years, kg |
63.6 ± 8.8 |
65.1 ± 11.7 |
0.38 |
|
≥3 kg of weight loss in the past, n (%) |
37 (51) |
44 (64) |
0.17 |
|
Current smoking, n (%) |
7 (10) |
10 (14) |
0.44 |
|
Current exercise habit, n (%) |
22 (31) |
21 (30) |
1.00 |
|
Pre-menopausal women, n (% of women) |
11 (58) |
11 (61) |
1.00 |
|
Working status |
|
|
|
|
Employed full-time, n (%) |
64 (89) |
60 (87) |
0.80 |
|
Shift or late-night work, n (%) |
2 (3) |
1 (1) |
1.00 |
|
Four-year college graduate or higher, n (%) |
52 (72) |
52 (75) |
|
|
Household income, n (%) |
|
|
0.71 |
|
<3,000,000 JPY |
8 (11) |
4 (6) |
|
|
3,000,000 JPY to 5,000,000 JPY |
10 (14) |
8 (12) |
|
|
5,000,000 JPY to 7,000,000 JPY |
14 (19) |
17 (25) |
|
|
7,000,000 JPY to 10,000,000 JPY |
21 (29) |
24 (35) |
|
|
≥10,000,000 JPY |
19 (26) |
16 (23) |
|
|
Living alone, n (%) |
14 (19) |
11 (16) |
0.66 |
|
Married, n (%) |
50 (69) |
54 (78) |
0.26 |
|
Body weight, kg |
76.5 ± 9.6 |
79.1 ± 11.3 |
0.15 |
- The biggest problem that I have is that the data doesn’t form a consistent narrative. In table 3 the drop in energy intake was almost 50% higher in the control group, 132 vs 85.5. On the other hand, the steps per day is almost 3 times that in the control group compared to intervention group. The activity data may be compromised as those participants that had less than 3 valid days of steps are very unlikely to be missing at random so it’s difficult to rely on this information. The primary result is that the loss in weight was more than 3 times greater in the intervention group 2.4 vs. 0.7, but how is this to be understood? It looks like those in the intervention group ate relatively more, exercised more and the exercise won out. The underlying problem with all of this is measurement fidelity. Have any studies been done to assess the reliability of the App? The other problem is the self-measurement. My understanding is that weight loss studies are notorious for participants (in the control group) not being scrupulously honest about their measurements. It may have been prudent to ask them to measure themselves at least once per month. My guess is that the authors need to revisit the data to better understand what is happening.
RESPONSE
We agree with your insightful comment on the measurement fidelity of our instruments. We acknowledge that our measurement tool for energy intake, a brief diet history questionnaire (BDHQ), has insufficient responsiveness to track changes in energy intake to dietary interventions. As discussed in lines 321–336, energy intake estimated from the dietary log decreased from 1833 kcal/day in the first two weeks to 1682 kcal/day in the final two weeks. A roughly 150 kcal/day drop in energy intake corresponds to approximately 1.9 kg weight reduction over 3 months. This calculation may explain in part the between-group difference in body weight. A recent study, authored in part by our research group members, demonstrated that the app’s automated image recognition system could estimate energy intake with acceptable validity (ref. 11). For the primary outcome, as we described in lines 193–196, we sent a weight scale (HD-665; Tanita, Tokyo, Japan) to each participant to measure their weight. We requested them to measure their weights in the morning after an overnight (≥10 h) fast and sent the data via self-service questionnaire software. We also prepared some types of rewards comparable to 20,000 JPY (money, gift card, healthcare device, or sporting goods) as described in lines 110–112. We expected these efforts contributed to high measurement fidelity. To treat our study with caution, we added the following sentence as a limitation in lines 372–374: Fourth, energy intake measured by BDHQ and physical activity measured by accelerometer could not fully explain the primary result of the body weight change.
Reference:
- Sasaki, Y., et al., Nutrient and Food Group Prediction as Orchestrated by an Automated Image Recognition System in a Smartphone App (CALO mama): Validation Study. JMIR Form Res, 2022. 6(1): p. e31875.
- The upshot is that this the results of this study as they stand have raised several questions that have not been addressed and these results must be interpreted very cautiously.
RESPONSE
We appreciate your constructive comments for improving the value of the current manuscript.

Reviewer 2 Report
As a statistician, this paper was a pleasure to read. Solid design, well-executed analysis.
One thing that could be added (but many don't...) would be a sentence or two regarding the underlying assumptions (goodness of fit, equal scatter...)
Author Response
We thank the reviewers for their constructive comments and appreciate the opportunity to revise the manuscript accordingly. Below is a point-by-point response to the concerns raised (reviewer comments are in italics), and in the manuscript, changes have been marked using the track-changes function.
Reviewer #2:
- As a statistician, this paper was a pleasure to read. Solid design, well-executed analysis.
RESPONSE
We appreciate your positive comments.
- One thing that could be added (but many don't...) would be a sentence or two regarding the underlying assumptions (goodness of fit, equal scatter...)
RESPONSE
In accordance with the comment, we checked the assumptions for the primary outcome, which is the change from baseline in body weight. The scatter is not significantly different between the intervention and control groups (p = 0.14 for homogeneity of variance). On the other hand, the normality for the primary outcome would be suspected (p < 0.001 for normality). Thus, we assessed the primary outcome using a non-parametric analysis with the Wilcoxon rank sum test. Figure R2 shows a significant difference between the intervention and control groups. We added this result in Table 2.
|
|
Intervention group |
Control group |
p (Wilcoxon) |
|
N |
71 |
69 |
|
|
Median [Q1, Q3] |
-1.30 [-3.15, -0.50] |
-0.40 [-1.60, 0.90] |
<0.001 |
|
Min, Max |
-27.00, 2.35 |
-21.80, 4.50 |
|
Figure R2. Non-parametric analysis and distribution for change from baseline in body weight
